# Translating the potential of the urine steroid metabolome to stage NAFLD (TrUSt-NAFLD): study protocol for a multicentre, prospective validation study

Hamish Miller [1,2] David Harman,[3] Guruprasad Padur Aithal,[4] Pinelopi Manousou,[5] Jeremy F Cobbold,[6,7] Richard Parker [8] David Sheridan,[9] Philip N Newsome,[10] Fredrik Karpe [1] Matthew Neville,[1] Wiebke Arlt,[11,12] Alice J Sitch,[10,13] Marta Korbonits,[14] Michael Biehl,[15,16] William Alazawi,[2] Jeremy W Tomlinson[1]

**Correspondence to**
Dr Jeremy W Tomlinson;
jeremy.tomlinson@ocdem.ox.ac.uk

## ABSTRACT

**Introduction** Non-alcoholic fatty liver disease (NAFLD) affects approximately one in four individuals and its prevalence continues to rise. The advanced stages of NAFLD with significant liver fibrosis are associated with adverse morbidity and mortality outcomes. Currently, liver biopsy remains the 'gold-standard' approach to stage NAFLD severity. Although generally well tolerated, liver biopsies are associated with significant complications, are resource intensive, costly, and sample only a very small area of the liver as well as requiring day case admission to a secondary care setting. As a result, there is a significant unmet need to develop non-invasive biomarkers that can accurately stage NAFLD and limit the need for liver biopsy. The aim of this study is to validate the use of the urine steroid metabolome as a strategy to stage NAFLD severity and to compare its performance against other non-invasive NAFLD biomarkers.

**Methods and analysis** The TrUSt-NAFLD study is a multicentre prospective test validation study aiming to recruit 310 patients with biopsy-proven and staged NAFLD across eight centres within the UK. 150 appropriately matched control patients without liver disease will be recruited through the Oxford Biobank. Blood and urine samples, alongside clinical data, will be collected from all participants. Urine samples will be analysed by liquid chromatography-tandem mass spectroscopy to quantify a panel of predefined steroid metabolites. A machine learning-based classifier, for example, Generalized Matrix Relevance Learning Vector Quantization that was trained on retrospective samples, will be applied to the prospective steroid metabolite data to determine its ability to identify those patients with advanced, as opposed to mild-moderate, liver fibrosis as a consequence of NAFLD.

**Ethics and dissemination** Research ethical approval was granted by West Midlands, Black Country Research Ethics Committee (REC reference: 21/WM/0177). A substantial amendment (TrUSt-NAFLD-SA1) was approved on 26 November 2021.

**Trial registration number** ISRCTN19370855.

## STRENGTHS AND LIMITATIONS OF THIS STUDY

⇒ The participants recruited with non-alcoholic fatty liver disease will all have had a liver biopsy to stage their disease severity as well as multiple other non-invasive biomarkers that are currently used in clinical practice.

⇒ The study includes a well-matched and phenotyped control cohort without evidence of liver disease, prospectively recruited from an existing biobank.

⇒ A urinary test has high patient acceptability and is less operator dependent than other non-invasive tests such as transient elastography.

⇒ The study is cross-sectional and not longitudinal, so will not be able to address questions as to the utility of biomarker strategies and disease progression/remission.

⇒ The study will recruit patients having a liver biopsy as part of routine clinical care which may introduce a degree of selection bias, creating a cohort more likely to have advanced (as opposed to mild) disease severity.

## INTRODUCTION

Non-alcoholic fatty liver disease (NAFLD) is the most prevalent chronic liver condition.[1] It affects up to 30% of the population (rising to 60% of patients with diabetes and obesity); >50 million adults in Europe and >10 million adults in the UK. It is the second most common indication of liver transplantation worldwide[2 3] and now accounts for 8.4% of liver transplants in Europe.[4] The advanced stages are associated with a fourfold to fivefold increase in mortality and morbidity, driven through liver-specific causes (including hepatocellular carcinoma (HCC)) and cardiovascular events.[5 6] Estimates suggest that NAFLD

costs the UK in excess of £5 billion per year and, globally, the expected 10-year burden of NAFLD could exceed $1.005 trillion in the USA and €334 billion in Europe.[7]

NAFLD is a spectrum of disease ranging from simple steatosis through to inflammation (non-alcoholic steatohepatitis) to fibrosis (graded from F0 to F4) with the risk of cirrhosis (F4) and the development of HCC. Fibrosis stage is tightly associated with an adverse clinical outcome with both liver-specific and cardiovascular morbidity and mortality.[8 9] The current reference standard for assessment of NAFLD severity is liver biopsy; however, there are significant complications and poor patient acceptability.[10 11] In addition, it is a resource-intensive and costly procedure, requires day case admission and time off work for the patient, samples a very small area of the liver and the interpretation of the biopsy findings is fraught with interindividual variability.[12] Therefore, the development of accurate, non-invasive markers to stage NAFLD is a critical unmet priority clinical need. Routine interpretation of liver biochemistry is unhelpful; 50% of patients with advanced fibrosis (F3 or F4) have normal liver chemistry.[13] Several non-invasive tools (serological, clinical, imaging) have been developed; however, to date, none have been shown to be sufficiently robust to replace liver biopsy.[14]

The liver is the main site of steroid metabolism and there is disruption of specific pathways in patients with NAFLD,[15 16] providing biological plausibility to our approach. The pilot study showed an increased activity in 11β-hydroxysteroid dehydrogenase type 1 and 5α reductase in patients with advanced fibrosis.[17] Steroid 5β reductase (AKR1D1), which is mostly expressed in the liver, has also been shown to decrease as fibrosis in NAFLD progresses.[16] When coupled with machine learning-based analysis using, for instance, Generalized Matrix Learning Vector Quantization (GMLVQ), a global assessment of hepatic steroid hormone metabolism can be made. GMLVQ is a supervised machine learning technique whereby an algorithm is trained using known datasets, and then subsequently applied to unknown samples. GMLVQ is able to simplify complex data into clusters of patients who have a high risk of having advanced fibrotic NAFLD. Data from our published proof-of-principle study indicate that this is tightly influenced by NAFLD stage.[17] As part of this discovery study, we analysed urine samples from 227 subjects (121 with biopsy-proven NAFLD and 106 controls). GMLVQ analysis achieved excellent separation of early (F0–F2) from advanced (F3–F4) fibrosis (area under the curve-receiver operating characteristic curve (AUC-ROC): 0.92 (0.91–0.94)). There was near perfect separation of healthy controls from patients with advanced fibrotic NAFLD and from those with NAFLD cirrhosis. Further refinement of the model incorporated the 10 most discriminatory steroids as well as patient age and body mass index (BMI) which has further enhanced diagnostic performance.[17]

The aim of the current study is to prospectively validate our published findings in an independent cohort of patients undergoing routine liver biopsy for NAFLD and to compare the performance of the machine learning-based urine steroid metabolome test against other non-invasive biomarker strategies that are currently used in clinical practice.

## METHODS AND ANALYSIS
### Design
TrUSt-NAFLD is a prospective, observational, non-interventional clinical study where recruitment and participation is designed to coincide with routine National Health Service (NHS) clinical care aiming to avoid the need for additional venepuncture, minimise inconvenience to participants and maximise recruitment potential. Occasionally, additional study-specific visits for participants will be scheduled where this can facilitate recruitment or ensure that samples are collected within the predefined time limitations of the study (samples collected within 12 months of liver biopsy).

A dedicated control cohort of participants without evidence of liver disease will be prospectively recruited from an existing biobank of patients (Oxford Biobank (OBB), https://www.oxfordbiobank.org.uk) who have undergone detailed metabolic phenotyping and provided consent to be recalled by phenotype for additional clinical studies. These participants will not undergo liver biopsies but will have a comprehensive assessment of non-invasive NAFLD biomarkers measured as well as urine steroid metabolite analysis.

### Setting and participants
The study will take place in secondary care hospital settings in eight dedicated hepatology clinics across the UK. Patients (male or female) aged ≥18 years, who are attending or have attended an NHS clinic for the diagnosis and/or management of NAFLD and who are scheduled to have a liver biopsy or who have had a liver biopsy to stage NAFLD within the last 12 months will be approached to participate. Control participants without evidence of NAFLD will be recruited from the OBB based at the Oxford Centre for Endocrinology, Diabetes and Metabolism, Churchill Hospital, Oxford, UK.

### Patient and public involvement
The study protocol and associated supporting documentation was reviewed by a dedicated patient and public involvement panel that is supported by the National Institute for Health and Care Research (NIHR) Oxford Biomedical Research Centre (BRC) and their feedback was incorporated into the final study documents. The panel will continue to be involved with dissemination activities following study completion.

### Recruitment
The TrUSt-NAFLD study will recruit patients with a diagnosis of NAFLD from dedicated NHS hepatology clinics. The initial approach for participants to be part of

the study will be made by members of the team directly involved in their clinical care. In some situations, an alternative recruitment strategy whereby patients (who have consented to be approached for future research) are contacted having been identified from existing clinical databases may be employed. To ensure sufficient recruitment of patients, eight centres across the UK will participate in the study. Interested potential participants will be seen during their routine NHS clinic visit by the research team to confirm eligibility. Those patients identified from existing databases will be contacted by letter and their eligibility will have been assessed prior to contact by lead clinicians and researchers within the dedicated hepatology team.

Recruitment of control participants without NAFLD will use the OBB; a biobank of >9000 volunteers in Oxfordshire, UK, who have undergone extensive metabolic phenotyping and consented to be reapproached for clinical research. In order to try to ensure the absence of significant liver disease, individuals with a Homeostatic Model Assessment for Insulin Resistance (HOMA-IR) (as a marker of global insulin resistance) (https://www.dtu.ox.ac.uk/homacalculator/) in the lowest 25%, BMI <35 kg/m$^2$ and fasting triglyceride levels <2 mmol/L will be invited to participate. HOMA-IR, triglycerides and BMI are all measured and recorded as part of the detailed OBB metabolic phenotyping and therefore can be used in a recall-by-phenotype strategy. Invitations to take part in the study will be sent with an aim of recruiting 60% male and 40% female participants to match the anticipated sex distribution in the NAFLD cohort which has been estimated from local liver biopsy data. Only individuals with normal liver biochemistry and non-invasive serum markers (including the enhanced liver fibrosis (ELF) panel) indicating low risk of advanced NAFLD will be included in the study.

### Screening and eligibility assessment

For potential participants with a diagnosis of NAFLD, screening will be undertaken by the clinical care team through direct confirmation of age, establishing that there is a diagnosis of NAFLD and that they either have a liver biopsy scheduled or have had a liver biopsy within the last 12 months. Potential participants will be provided with a letter of invitation and a patient information leaflet (PIL) either at the time of their routine NHS clinic visit or sent by post. For potential control participants without NAFLD, screening of the existing OBB database will be undertaken and a letter of invitation and PIL will be sent by members of the OBB research team.

Specific inclusion and exclusion criteria are detailed below. There will be no exceptions made regarding eligibility and each participant must satisfy all the approved inclusion and exclusion criteria of the protocol.

### Inclusion criteria
#### Participants with NAFLD
▶ Participant is willing and able to give informed consent for participation in the study.
▶ Patients with a diagnosis of NAFLD who are scheduled for a liver biopsy, or the patient has had a liver biopsy with a confirmed diagnosis of NAFLD within the last 12 months.
▶ Aged ≥18 years.

#### Control participants without evidence of NAFLD
▶ Participant is willing and able to give informed consent for participation in the study.
▶ Bottom 25th percentile of HOMA-IR in OBB.
▶ BMI <35 kg/m$^2$.
▶ Fasting triglycerides <2.0 mmol/L.
▶ Aged ≥18 years.

### Exclusion criteria
#### All participants
▶ Insufficient understanding of written and verbal English.
▶ The participant has not had sufficient time to read the PIL and understand the study requirements.
▶ Hepatic steatosis, inflammation or fibrosis with a primary aetiology other than NAFLD.

### Study procedures
All participants will personally sign and date the latest approved version of the informed consent form before any study-specific procedures are performed. Written and verbal versions of the PIL and informed consent forms will be presented to the participants detailing the exact nature of the study, what it will involve for the participant, the implications and constraints of the protocol and any risks involved in taking part. It will be clearly stated that the participant is free to withdraw from the study at any time for any reason without prejudice to their future care, without affecting their legal rights and with no obligation to give the reason for withdrawal. The participant will be allowed as much time as possible (and this may be less than 24 hours) to consider the information, and the opportunity to question the investigators, their general practitioner or other independent parties to decide whether they will participate in the study. Consent may be obtained, and the study procedures performed on the same day. The person who obtains the informed consent will be suitably qualified, experienced and have been authorised to do so by the chief/principal investigator. A copy of the signed informed consent form will be given to the participant. The original signed form will be retained at the study site. The study is not randomised. Participants will be enrolled as they are consented. At enrolment, all participants will be assigned a unique identification number in a sequential order.

For all participants, demographic data will be collected including age, sex, ethnicity, height, weight and calculated BMI. Details of the participants' clinical history will

also be collected, including any relevant medical history (smoking, type 2 diabetes, hypertension, cardiovascular disease and alcohol use) as well as details of concomitant prescribed medications. Patients with NAFLD will usually have routine blood tests as part of their clinical care consultation and therefore the additional blood samples that will be collected will not require additional venepuncture (although this may be needed if a participant attends for a dedicated research visit that is separate from their clinical care appointment). A dedicated research visit is required for all participants without NAFLD that will include collecting demographic data and clinical history as well as venepuncture and providing a 10 mL urine sample. Vibration-controlled transient elastography (VCTE) is frequently used in hepatology clinics to determine the risk of advanced fibrosis and can provide both a measure of liver stiffness (indicative of fibrosis stage) and a controlled attenuation parameter that correlates with the degree of hepatic steatosis.[18 19] Where available, these data will be recorded for all patients with NAFLD.

All blood sample analyses will include measurement of a full blood count, renal function and electrolytes, liver chemistry (including alanine aminotransaminase (ALT), aspartate aminotransaminase (AST), alkaline phosphatase, gamma-glutamyltransferase, bilirubin, albumin, ferritin), total and high-density lipoprotein-cholesterol and triglycerides as well as glycated haemoglobin. In addition, samples will be analysed for the ELF panel[20] (hyaluronic acid, type III procollagen peptide and tissue inhibitor of matrix metalloproteinase 1). Calculations for non-invasive risk score to detect advanced fibrosis will also be made including NAFLD Fibrosis Score (NFS), Fib-4, AST to Platelet Ratio Index (APRI), Fibroscan-AST (FAST) and BMI, AST/ALT Ratio, Diabetes Mellitus (BARD) .[21–25]

Liver biopsy data will be collected to include the NAFLD Activity Score grading with its individual components of steatosis, lobular inflammation, and ballooning and also the fibrosis staging (graded F0–F4) according to the Kleiner classification.[26]

A 10 mL urine sample will be collected from all participants, aliquoted and stored at −80°C until analysis. Predefined steroid hormone metabolites will be measured using gas chromatography-mass spectrometry and liquid chromatography-tandem mass spectrometry (LC-MS-MS) as described previously.[27 28] The steroid metabolite data will then be subjected to machine learning employing a GMLVQ[17] classifier that was trained on retrospective data. Its performance to identify participants with advanced fibrotic NAFLD compared against other non-invasive biomarker strategies (Fib-4, NFS, BARD, APRI, AST/ALT, ELF, FAST, VCTE) will be evaluated.

### Statistical methods, sample size and analysis plan

The study has been planned and designed with expert statistical advice and an experienced statistician is part of the investigative team.

### Sample size

We will aim to recruit at least 460 participants to the TrUSt-NAFLD study (NAFLD n=310; control n=150). Sample size and power calculations have been based on our previous published data.[17] Our primary analysis will aim to distinguish patients with mild (F0–F2) versus advanced fibrosis (F3–F4). Through the recruitment of 310 participants with NAFLD, we will have 85% power to identify a difference in sensitivity between technologies of 12 percentage points (80% vs 92%) assuming a 5% significance level, a prevalence of 50% (ie, 50% of the participants will be F0–F2 and 50% F3–F4) and independence of comparative test results. The estimates of sensitivity and the difference in sensitivity have been estimated from previous work.[14]

### Analysis of outcome measures

Data for all participants who complete the study will be included in the analysis. If any participants withdraw from the study, samples and data collected will not routinely be destroyed and will be used in the analysis unless requested not to do so by the individual. Statistical analysis will be undertaken at the end of study and after the analysis of all the samples has been completed.

The primary outcome of the TrUSt-NAFLD study is to determine the accuracy of the urine steroid metabolome GMLVQ algorithm to distinguish early (F0-2) from advanced (F3–F4) NAFLD fibrosis. Sensitivity, specificity and negative and positive predictive values will be calculated with 95% CIs. Additionally, ROC curves will be generated and the AUC will be calculated.

Secondary and exploratory outcomes will include a comparison of the performance of the urine steroid metabolome GMLVQ algorithm against other non-invasive NAFLD biomarkers (Fib-4, NFS, BARD, APRI, AST/ALT, ELF, FAST, VCTE). Data from participants with NAFLD will also be compared against control participants without evidence of NAFLD to determine the ability of the algorithm to detect NAFLD (across the spectrum of its severity) in the general population. Finally, exploratory analyses detecting steroid metabolites using LC-MS-MS in serum samples and subsequently incorporating the GMLVQ algorithm will be performed and compared against the urine analysis.

### DISCUSSION

This diagnostic test using the urinary steroid metabolome represents a unique way to diagnose and stage NAFLD without the need for blood tests, biopsy or imaging. It has shown promising results in a published pilot study and the TrUSt-NAFLD study aims to validate these findings in a larger prospectively collected cohort. Urine steroid metabolites are measured using LC-MS-MS before applying machine learning techniques including GMLVQ. During the pilot study, prototype urine steroid metabolome profiles for each fibrosis stage were created using GMLVQ. When the algorithm is presented with an

unknown sample, the dissimilarity between the unknown urine steroid profile and the profile of established prototype is measured. Based on this, the sample is then assigned a disease stage based on the closest similarity to a specific prototype. Both the LC-MS-MS and machine learning analysis represent potential drawbacks of this study given their relative complexity. In addition, such an analytical capability may not be currently available at all clinical centres. However, LC-MS-MS is a relatively high-throughput approach where samples can be analysed rapidly. A strength of the study is that the participants with NAFLD have biopsy-proven and histologically staged disease. However, within the control cohort, it is neither realistic nor ethically acceptable to undertake a liver biopsy (bearing in mind the associated morbidity and mortality), but this serves to emphasise the importance of developing non-invasive tests with good accuracy alongside high patient acceptability. Should the data from this study validate our approach, then there is the realistic potential for major clinical impact, changing how NAFLD is diagnosed and staged.

## ETHICS AND DISSEMINATION
### Ethical and safety considerations
The chief investigator will ensure that this study is conducted in accordance with the principles of the Declaration of Helsinki and that the study is conducted in accordance with relevant regulations and with Good Clinical Practice.

The study will not pose significant ethical issues or risks to the participants. Sample collection and study visits for those individuals with NAFLD will, in the majority of cases, coincide with routine clinical care appointments, and so additional venepuncture, and its associated discomfort, will not be needed. Providing the urine sample provides no additional risk. Where additional study appointments are required, travel expenses are reimbursed and a small payment to cover inconvenience will be offered. Incidental abnormal findings unrelated to the study will be discussed with the participants and where appropriate, and with their consent, referral to their general practitioner or suitable specialist will be made.

The TrUSt-NAFLD study will be sponsored by the University of Oxford, UK. All relevant studies supporting documentation as well as the clinical protocol have been reviewed and approved by West Midlands, Black Country Research Ethics Committee (REC reference: 21/WM/0177) and the Health Research Authority (HRA) (https://www.hra.nhs.uk).

The OBB is sponsored by the University of Oxford, UK, and supported by the NIHR Oxford BRC. The protocols and supporting documentation for recruitment, investigations and recruiting by phenotype and genotype have been reviewed and approved by South Central, Oxford C NHS REC (ref 18/SC/0588) and the HRA. An annual progress report will be submitted to the REC and HRA (where required) host organisation and sponsor. In addition, on study completion, an end of study notification and final report will be submitted to the same parties.

### Transparency in research
Details of the study will be made available on HRA research summaries (https://www.hra.nhs.uk/planning-and-improving-research/application-summaries/research-summaries/). The study details will also be available on University of Oxford and Radcliffe Department of Medicine websites. The study is registered on the International Standard Registered Clinical/Social Study Number (ISRCTN) database (ISRCTN19370855, https://doi.org/10.1186/ISRCTN19370855).

### Data statement
The datasets generated and/or analysed during the current study will be stored in a non-publicly available repository.

### Dissemination plan (publications, data deposition and curation)
A copy of the final report and/or publication will be sent to study participants on request. Results will be disseminated through presentations at national and international conferences and publications sent to peer-reviewed journals with open access in accordance with the funder's publication policy (https://wellcome.org/grant-funding/guidance/open-access-guidance/open-access-policy). In addition, research findings will be disseminated through dedicated departmental public engagement events and through patient support groups.

The investigators will be involved in reviewing drafts of the manuscripts, abstracts, press releases and any other publications arising from the study. Authors will acknowledge that the study was funded by Wellcome. Authorship will be determined in accordance with the ICMJE guidelines and other contributors will be acknowledged.

**Author affiliations**
[1]Oxford Center for Diabetes, Endocrinology and Metabolism, University of Oxford, Oxford, UK
[2]Barts Liver Centre, Queen Mary University London and Barts Health NHS Trust, London, UK
[3]Royal Berkshire Hospital NHS Foundation Trust, Reading, UK
[4]NIHR Nottingham Biomedical Research Centre, Nottingham University Hospitals NHS Trust and University of Nottingham, Nottingham, UK
[5]Department of Metabolism, Digestion and Reproduction, Imperial College London, London, UK
[6]Oxford Liver Unit, Oxford University Hospitals NHS Foundation Trust, Oxford, UK
[7]NIHR Oxford Biomedical Research Centre, Oxford University, Oxford, UK
[8]Leeds Liver Unit, Leeds Teaching Hospitals NHS Trust, Leeds, UK
[9]Institute of Translational and Stratified Medicine, University of Plymouth, Plymouth, UK
[10]National Institute for Health Research Birmingham Biomedical Research Centre, University Hospitals Birmingham NHS Foundation Trust and the University of Birmingham, Birmingham, UK
[11]Institute of Metabolism and Systems Research, University of Birmingham, Birmingham, UK
[12]Medical Research Council London Institute of Medical Sciences, Imperial College London, Hammersmith Campus, London, UK
[13]Institute of Applied Health Research, University of Birmingham, Birmingham, UK

[14]Centre for Endocrinology, William Harvey Research Institute, Barts and the London School of Medicine and Dentistry, Queen Mary University of London, London, UK
[15]Faculty of Science and Engineering, Bernoulli Institute for Mathematics, Computer Science and Artificial Intelligence, University of Groningen, Groningen, Netherlands
[16]SMQB, Institute of Metabolism and Systems Research, University of Birmingham, Birmingham, UK

**Contributors** HM will coordinate and run the TrUSt-NAFLD study and drafted the manuscript. JWT is the chief investigator for the study and was responsible for the original concept and design of the study. JWT edited the manuscript and was responsible for the final approval of the manuscript. DH, GPA, PM, JFC, RP, DS, PNN, FK, MN, WA, AJS, MK, MB, WA and JWT were all involved in securing funding; they have all reviewed the protocol and are named investigators and they have all contributed to the writing and reviewing of the manuscript.

**Funding** This study is funded through a Wellcome Innovator Award to JWT, MB and WA (ref: 222627/Z/21/Z) and is supported by the NIHR Oxford Biomedical Research Centre and by the NIHR Birmingham Biomedical Research Centre.

**Disclaimer** The views expressed are those of the authors and not necessarily those of the NIHR or the Department of Health and Social Care.

**Competing interests** JWT has been an advisory board member for Novo Nordisk, Pfizer and Poxel and has been part of a data and safety monitoring committee for Novartis. RP has been an advisory board member for Novo Nordisk.

**Patient and public involvement** Patients and/or the public were involved in the design, or conduct, or reporting, or dissemination plans of this research. Refer to the Methods section for further details.

**Patient consent for publication** Not applicable.

**Provenance and peer review** Not commissioned; externally peer reviewed.

**ORCID iDs**
Hamish Miller http://orcid.org/0000-0002-1198-6093
Richard Parker http://orcid.org/0000-0003-4888-8670
Fredrik Karpe http://orcid.org/0000-0002-2751-1770

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
