## [Reviewer comments · BMJ Open]

ARTICLE DETAILS

TITLE (PROVISIONAL)	Translating the potential of the urine steroid metabolome to stage NAFLD (TrUSt NAFLD) – study protocol for a multi-centre, prospective validation study
AUTHORS	Miller, Hamish; Harman, David; Aithal, Guruprasad; Manousou, Pinelopi; Cobbold, Jeremy F.; Parker, Richard; Sheridan, David; Newsome, Philip; Karpe, Fredrik; Neville, Matthew; Art, Wiebke; Sitch, Alice; Korbonits, Marta; Biehl, Michael; Alazawi, William; Tomlinson, Jeremy

VERSION 1 – REVIEW

REVIEWER	Wilson, Thomas Aberystwyth University, Life Sciences
REVIEW RETURNED	01-Aug-2023

GENERAL COMMENTS	The submitted protocol is excellently presented, and details a protocol that is both scientifically robust and has the potential to significantly accelerate the translation and further use of non-invasive biomarkers in clinical care. The authors have been thorough in their presentation of this protocol and it is my recommendation that this manuscript is acceptable for publication without revision. I look forward to reading the results of this study in the future.
---

REVIEWER	Liu, Fei Harvard Medical School, Medicine
REVIEW RETURNED	04-Nov-2023

GENERAL COMMENTS	The passage lacks discussion of potential challenges or drawbacks in using the urine steroid metabolome as a diagnostic tool, which would provide a more balanced perspective. Additionally, no solution is presented for the bias created by only enrolling NAFLD patients. The study's complexity, particularly its use of machine learning, may require a simpler explanation for non-experts in the field. Finally, some technical terms and abbreviations (such as GMLVQ) are introduced without sufficient explanation, potentially making it less accessible to a wider audience.
--

VERSION 1 – AUTHOR RESPONSE

Reviewer 1 general comment:

COMMENT:

The submitted protocol is excellently presented, and details a protocol that is both scientifically robust and has the potential to significantly accelerate the translation and further use of non-invasive biomarkers in clinical care. The authors have been thorough in their presentation of this protocol and it is my recommendation that this manuscript is acceptable for publication without revision. I look forward to reading the results of this study in the future.

ANSWER: We thank the reviewer for their positive comments about the manuscript.

Reviewer 2 general comment:

COMMENT: The passage lacks discussion of potential challenges or drawbacks in using the urine steroid metabolome as a diagnostic tool, which would provide a more balanced perspective. Additionally, no solution is presented for the bias created by only enrolling NAFLD patients. The study's complexity, particularly its use of machine learning, may require a simpler explanation for non-experts in the field. Finally, some technical terms and abbreviations (such as GMLVQ) are introduced without sufficient explanation, potentially making it less accessible to a wider audience.

	Reviewer' specific comments	Author response and changes made	Page and line numbers in revised paper where the change can be found
1	The passage lacks discussion of potential challenges or drawbacks in using the urine steroid metabolome as a diagnostic tool, which would provide a more balanced perspective.	We thank the reviewer for drawing this to our attention and we have therefore added a discussion section.	Page 13, lines 388-412
2	Additionally, no solution is presented for the bias created by only enrolling NAFLD patients.	We are sorry that we did not make this clearer in the manuscript, but we are recruiting 150 participants without NAFLD against which we will compare our NAFLD cohort.	Page 7, lines 208-213
3	The study's complexity, particularly its use of machine learning, may require a simpler explanation for non-experts in the field.	We are grateful for this valid point and have endeavoured to clarify the complexities of the approaches used within the study, especially with regards machine learning. This is now included within the dedicated discussion section.	Page 13, lines 388-412

4	Finally, some technical terms and abbreviations (such as GMLVQ) are introduced without sufficient explanation, potentially making it less accessible to a wider audience.	We thank the reviewer for this comment and we have therefore added some explanations and expanded the abbreviations for the more complex aspects of the study, including GMLVQ.	Page 6, lines 178-180. Page 7, lines 181
---	---	---	---

VERSION 2 – REVIEW

REVIEWER	Wilson, Thomas Aberystwyth University, Life Sciences
REVIEW RETURNED	29-Nov-2023
GENERAL COMMENTS	The authors have addressed all of the reviewer's and editors's comments. This manuscript should be accepted for publication without further revision.

VERSION 2 – AUTHOR RESPONSE